# Transparency Index of the Supplying Countries' Institutions and Tree Cover Loss: Determining Factors of EU Timber Imports?

**Encarnación Moral-Pajares [1,\*], Concepción Martínez-Alcalá [1], Leticia Gallego-Valero [1,\*] and Ángela Andrea Caviedes-Conde [2]**

[1] Department of Economy, University of Jaén, 23071 Jaén, Spain; cmalcala@ujaen.es

[2] Department of Applied Economics II, Rey Juan Carlos University, 28032 Madrid, Spain; andrea.caviedes@urjc.es

[\*] Correspondence: emoral@ujaen.es (E.M.-P.); lgallego@ujaen.es (L.G.-V.)

**Abstract:** Illegal logging and the associated deforestation have serious consequences for biodiversity, the climate, the economy and society. The EU Timber Regulation (EUTR) prohibits the placing of illegally harvested timber or timber products on the market. The objective of this paper is to analyse the recent evolution of EU imports of these products from the international market, in order to check how the transparency index of the supplying countries' institutions and tree cover loss have influenced this trajectory. To that end, a panel data model is estimated with 228 observations from 38 exporting countries between 2012 and 2017. The results show that EU timber imports have a direct association with the transparency index and an inverse relationship with tree cover loss; both these relationships are highly significant at the one-percent level. Other significant factors are the performance of the EU construction sector (as a proxy for timber demand) and timber supply. In the short and medium-term, Voluntary Partnership Agreements (VPAs) signed between the EU and non-EU timber-producing countries have a negative influence on the supply to EU member states. This study presents an analysis of EU timber imports after the implementation of the EUTR, providing specific conclusions that can inform policymakers' efforts to foster sustainable forest management.

**Keywords:** EU; timber; imports; EU Timber Regulation (EUTR); tree cover loss; forest certification

## 1. Introduction

Global concerns about the impacts of illegal logging and trade in timber in terms of deforestation and forest degradation prompted the European Commission to adopt the EU Action Plan for Forest Law Enforcement, Governance and Trade (FLEGT) in 2003 [1,2]. The EU is one of the world's largest importers of timber, and its actions can play a role in improving the legality of the production and sale of timber. In 2017, EU countries accounted for more than 35% of total world imports of timber and timber products, and, although almost three-quarters of these trade flows are within the EU [3], purchases from third countries represented 26.35% of the total. Furthermore, part of this trade between EU members is in timber or timber products initially purchased from outside the EU [4]. Between 2012 and 2017, EU imports from non-EU countries registered a cumulative annual growth rate of 5.23%, while those from partner countries recorded a growth rate of 4.73%.

The FLEGT establishes a package of measures to simultaneously influence both the demand and the supply of legally produced timber. The FLEGT represents the EU's first tool for improving forest management and encouraging the trade of legally produced timber [5]. To achieve this objective,

Regulation (EU) 2173/2005 and Regulation (EU) 1024/2008 authorised the European Commission to sign Voluntary Partnership Agreements (VPAs) with non-EU countries and implemented timber legality assurance systems in timber-producing countries. VPAs should be developed through an inclusive governance process involving stakeholders and should identify measures to control the illegal production and trade of export timber [6]. The EU subsequently adopted Regulation 995/2010 (the EU Timber Regulation or EUTR), prohibiting the placing of illegally harvested timber and timber products on the EU market [7] and implementing Regulation 607/2012. Such regulations require operators placing timber and timber products on the European market for the first time to check the legality of the imports [8] through a due diligence system (DDS) and stipulates the traceability of timber. The abovementioned legislation, which complements the VPAs, recognises the FLEGT licences as sufficient proof of the legal origin of timber [9].

There is compliance with the regulation entails costs for EU companies of varying amounts, depending on the subsector in question, as has been estimated by the European Commission [10]. Similarly, the application of VPAs and the FLEGT licensing system, like any other certification process, entails a financial cost to the country implementing them, which is partly covered by funding from EU projects and requires the development of capabilities to ensure the reliability and credibility of the process [11]. In return, products certified with the FLEGT licence gain automatic access to the EU market [12]. Public and private contracting policies increasingly stipulate the use of legal timber and the exclusion of illegal or unidentified timber [2,13]. Companies in the timber sector that take part in public tenders—for example, bidding to provide the timber needed for buildings or different types of construction works—must provide proof of the legality of the product offered. According to the European Commission, the trade measures derived from the EUTR are aimed at combating illegal timber production based on bilateral and voluntary agreements and, therefore, do not pose compatibility problems with the World Trade Organization [14]. These measures are targeted at very precise objectives, applying to individual shipments rather than at company or country level. The aim is to prevent illegal trade flows instead of legitimising them. Moreover, the intention is not to impede exports from less-developed countries to higher income markets but to ensure stricter standards for the protection of the natural environment and forest land [15].

The purpose of this research is two-fold. First, it analyses the recent evolution of EU imports of timber and timber products from the international market and identifies geographical patterns (from 2012, the year in which preparations were being made for the imminent entry into the force of the EUTR and following the effects triggered by the 2008 economic crisis). Second, it examines how certain variables, in-line with the studies of Hurmekoski et al., Paluš et al., Zhang et al. and Rougieux and Damette [16–19], have conditioned the recent evolution of these imports. Specifically, the analysis focuses on the trade flows of products included under the tariff headings listed in Chapter 44, timber and timber products, as set out in the Combined Nomenclature established in Council Regulation (EEC) 2658/87, summarised in Annex I of the EUTR. However, other products included in this annex (those in Chapters 47 and 48—pulp—and in Chapter 94—wooden furniture and prefabricated buildings —) are not considered. They are classified in other chapters of the tariff system and have significant differences from the products in Chapter 44, which affects trade patterns and complicates the analysis. The documentary and statistical information used is sourced from the European Commission, UN COMTRADE, EUROSTAT, FAOSTAT, Global Forest Watch, Transparency International and Google maps. This research analyses the influence on cross-country trade flows of international regulations that seek to promote the legality of the timber trade. Specifically, this paper enables an approximation of the effect of the EUTR on EU imports of timber and timber products in recent years, drawing specific conclusions that can guide policymakers' efforts to foster sustainable forest management.

Previous studies, such as that by Prestemon [20], have analysed how the regulations implemented by national governments have affected the price and demand for timber in the domestic market. Similarly, Giurca et al. [21], Jonsson et al. [22], Masiero et al. [23] and Pepke et al. [24] have analysed the influence of the EUTR on the intensity of EU imports and the characteristics of these imports in terms of product types and countries of origin, even before it came into effect. This paper

seeks to verify the possible link between EU-28 countries' imports of timber and timber products from third countries and various economic and institutional variables. The results of this research contribute to the literature that analyses how international standards aimed at promoting the sustainability of forests affect trade flows between countries. Furthermore, the findings offer an understanding of patterns in imports of timber and timber products in the years following the implementation of the EUTR. By performing a rigorous analysis of the available empirical data, this article seeks to contribute to the knowledge about EU countries' purchases of timber and timber products from non-EU countries. Moreover, the specific conclusions drawn can guide policymakers' actions towards achieving sustainable forest management.

This paper is structured in six sections, including this introduction. The next section contains theoretical arguments and hypotheses to be tested. The third section presents the materials and methods. The fourth section details the results, including a descriptive analysis of the timber trade flows of the EU countries between 2012—one year before the planned entry into the force of the EUTR—and 2017, the last year with available information, and a panel data analysis. The fifth section includes the discussions of the study, and lastly, the sixth section draws conclusions and provides some suggestions.

## 2. Theoretical Arguments and Hypotheses to Be Tested

The EU's commitment to tackling illegal timber means requires companies placing timber and timber products on the market to adopt systems of due diligence that preclude illegal products, basing their risk analyses not only on documentary checks but, also, verifying factual reports by the European Commission produced by third parties [25]. Given these requirements, as Borsky et al. [26] state, trade may be diverted towards countries with sound institutions capable of minimising the risk of illegal logging. In parallel, there is likely to be a reduction in imports from countries with less rigorous forest management policies. In light of the above, we can formulate the following hypothesis:

Hypothesis 1 (H1). The requirements established by the EUTR for the import of legal timber have promoted EU imports of timber from countries that have highly transparent government institutions.

The measures proposed by the Commission to mitigate the risk associated with the import of illegal timber [27] include drawing on data from the monitoring carried out by public and private entities in order to gain objective information about forest activity in the country where the supplier is located. In this regard, the percentage of tree cover loss may be indicative of the type of governance and national policy applied to tackle deforestation and forest degradation. According to the studies by Park [28] and Youn and Vadell et al. [29], governments' responsiveness in terms of protecting forests may determine the implementation of actions aimed at ensuring the conservation of this natural resource, investing in initiatives to prevent tree cover loss. This argument lays the foundation for the following hypothesis:

Hypothesis 2 (H2). Greater tree cover loss in the exporting country negatively affects EU timber imports.

In opening up to trade, national economies have an incentive to access and adopt new, greener production formulas [30]. When a country is integrated into the world economy, its export sector is more exposed to the legal requirements to ensure environmental protection imposed by the main importers. According to Obidzinski et al. [31] and Nathan et al. [32], producers that do not operate in accordance with these standards—mostly small-scale producers—are expelled from the market. Market-based instruments, such as taxes and command control tools (emission or exploitation limits) trigger an innovation effect in companies, as Ambec el al. point out [33]. More and more companies and consumers are concerned about their suppliers' compliance with the regulations established by governments to ensure the conservation of natural resources. Therefore, as Tricatollis et al. [11] and Lanoie et al. [34] argue, producers are more likely to be chosen as a trade partner when they promote legal practices to guarantee sustainability. Specifically, as Borsky et al. [26] and Rodriguez and Soumonni [35] claim in their studies of the tropical timber sector, product sustainability is a significant determinant of exports.

The EUTR requires operators (the companies that first place timber on the EU market, i.e., importers) to exercise due diligence to ensure that the timber is completely legal, assessing the use of third countries' independent verification systems and whether they conform to the provisions of the Regulation. This is the case of the Forest Stewardship Council (FSC) and the Program for the Endorsement of Forest Certification (PEFC), recognised as risk assessment and mitigation elements of due diligence systems, as stated by Holopainen et al. [36] and Halalisan et al. [37]. According to Nussbaum and Simula [38], forest certification is a voluntary process whereby an independent third party assesses the quality of forest management and production with respect to requirements established by independent certification organisations. As established by Tricatollis et al. [11], Lewis [39], Cubbage et al. [40], Kusonyola et al. [41] and Dias et al. [42], in various different cases, the forest certification systems established have helped to halt deforestation, restore natural ecosystems and conserve biodiversity. Given these considerations, the following hypothesis can be formulated:

Hypothesis 3 (H3). There is a positive relationship between the number of forest hectares with FSC or PEFC certification in exporting countries and EU timber imports.

The analysis also includes a gravity approach applied to the value of imports. Due to its explanatory power, the gravity model has been extensively used to analyse trade flows between countries. Tinbergen [43] first introduced the traditional gravity model to explain bilateral trade flows with no discriminatory trade barriers. Inspired by Newton's Law of Gravitation, there are three main explanatory variables in the gravity model of trade: the productive capacity of the exporting country, the demand of the importing country and the cost of transportation, which introduces the effect of distance between partners, as established by Anderson [44]. According to Paluš et al. [17], it is expected that the trade flow is positively related to the production and consumption levels of the trading countries and inversely linked to the distance between them. Subsequent empirical studies have introduced other variables in addition to those included in the simpler gravity model; for example, the influence of artificial barriers to international trade has been analysed. In this regard, the studies by Yang and Martínez-Zarzoso [45] and Carrère [46] mainly account for trade agreements between countries, including trade preferences, which can facilitate bilateral trade with partner countries

The gravity model has been used by Chan and Au [47], Thi Thu Thuong [48], Natale et al. [49] and Morley et al. [50] to analyse international trade flows in a wide variety of manufactured products, agri-food products and seafood, as well as in the field of services. It has even been used in a number of studies on the subject of the present research—namely, following Borsky et al. [26] and Houghton and Naughton [51], the international timber trade specifically for the case of tropical timber. It has also been applied by Akyüz et al. [52] to study the trade in forest products between the EU and Turkey and by Buongiorno et al. [53] to analyse the world trade in forest products. More recently, a structural model by Morland et al. [54] has been developed that examines trade flows for 13 types of forest sector products. Taking into account the characteristics of the sector analysed and the results of the research by Hurmekoski et al. [16] and Manninen [55], it has been decided that the best proxy for the evolution of the EU timber demand is the trajectory of its construction sector. On the other hand, following O'Brien and Bringezu [56] and Morland et al. [57], the production capacity of the exporting country is linked to the hectares of commercial forest land, with the quantity of available resources as a proxy for timber supply in the international market.

Based on the arguments set out above, this paper examines whether there is a relationship between purchases in the international market and the transparency of the supplying countries' institutions, the tree cover loss in these countries, the number of forest hectares in the supplier country in which forest management is certified by an independent third party in accordance with an internationally recognised scheme, the development of the construction sector in the EU (the main determinant of demand for intermediate timber products), the productive forest hectares in the country (timber supply), the average distance the imported products have to travel and whether there is an agreement to promote legal timber trade between partners. Specifically, a Voluntary Partnership Agreement (VPA) signed with the EU-28 or a bilateral coordination mechanism for forest law compliance (BCM), as is the case of China, signed in 2009.

## 3. Materials and Methods

The aim of the analysis is to identify the possible determinants of EU imports of timber and timber products from non-EU countries in recent years, following the implementation of the EUTR. To that end, we conducted a detailed analysis of the relevant variables and tested the relationship between the independent variables and the dependent variable. The data on EU imports of timber and timber products were gathered from the United Nations Commodity Trade Statistics Database (UN COMTRADE), where commodity groups are recorded according to the Harmonized Commodity Description and Coding System (Harmonized System or HS), or extended versions based on the HS, such as the Combined Nomenclature (CN) used by EU member countries, and are expressed in thousands of dollars [3]. The information about the independent variables, which comes from official statistics, is, in many cases, incomplete or based on estimates; as such, it may yield biased measures of the real situation. In particular, there are gaps for certain variables and countries. Our final sample includes data from 38 economies for the period 2012–2017. A panel data analysis was carried out [58]. The following clarifications should be taken into account:

The non-EU countries included in the analysis are Russia, China, the USA, Ukraine, Norway, Brazil, Canada, Belarus, Indonesia, Switzerland, Malaysia, Cameroon, Bosnia-Herzegovina, Gabon, Chile, Serbia, Uruguay, Côte d'Ivoire, the Congo, Turkey, Democratic Republic of the Congo, Ghana, New Zealand, Vietnam, South Africa, Myanmar, India, Ecuador, Thailand, Morocco, Peru, Equatorial Guinea, Australia, Bolivia, Albania, Central African Republic, Suriname and Liberia. Together, these countries account for more than 98% of all EU imports of timber and timber products between 2012—once the world timber market recovered after the 2008 crisis, and the year in which preparations were being made for Regulation 607/2012—and 2017, the last year for which statistical information is available for all the variables considered.

(a) The sample contains data for 6 consecutive years, covering the period between 2012—the year prior to the adoption of the EUTR regulation, in which operators in the sector prepared for the change and certain agents acted in anticipation of the effects of this regulation [24]—and 2017, the last year for which data are available.

(b) The dependent variable is the volume of EU imports of the timber and timber products included in Chapter 44 of the Combined Nomenclature, as set out in the Annex to the EUTR. Specifically, the following headings and subheadings are included in the analysis: 4401 (fuel wood); 4403 (wood in the rough); 4406 (sleepers); 4407 (sawn or chipped wood); 4408 (sheets for veneering); 4409 (wood); 4410 (particle board); 4411 (fibre board); 4412 (plywood); 4413 00 00 (densified wood); 4414 00 (wooden frames); 4415 (packing cases, crates, etc.); 4416 00 00 (casks and barrels) and 4418 (builders' joinery and carpentry of wood) [3].

(c) In-line with the literature reviewed in the previous section, the model includes a total of 7 independent variables; the description, source and expected sign of each variable are presented in Table 1. The relationships between these variables are proposed in the hypotheses set out in the Section 2.

(d) Limitations. The study only focuses on the products included in Chapter 44 and does not account for the imports of products included in Chapter 94. Likewise, the paper focuses on a series of variables; however, it does not consider certain factors that may have an influence on the evolution of timber imports by EU countries. For instance, variables that could be further examined in future research include indirect imports through third countries in which regulations governing legality are less stringent, EU-28 investments/divestments in producer countries' timber industries or the trade flows of timber from China and the USA, the main players in the world timber market.

**Table 1.** Independent variables used in the analysis: description, sources and expected relationship with the dependent variable.

| Variable | Description | Source | Expected Sign |
|---|---|---|---|
| Transparency Index (CPI) | Proxy used: Corruption Perceptions Index | Transparency International | + |
| Tree cover loss (TCL) | Proxy used: tree cover loss percentage | Global Forest Watch | − |
| Certified forest hectares (CFH) | Number of hectares certified by FSC and PEFC | FAOSTAT | + |
| EU Timber demand (TD) | Proxy: EU construction sector Gross Value Added | EUROSTAT | + |
| Timber supply of the country i (TS) | Productive forest hectares in country i | FAOSTAT | + |
| Distance (D) | Distance in km between the capital of country i and Brussels (considered the centre of the EU and where the main institutions are located) | Google maps | − |
| Voluntary Partnership Agreement or similar * (Ag) | Dummy: Agreement on timber trade with the EU | European Parliament | + |

Note: The variables refer to the country of origin of timber imports (Chapter 44) of the EU. * The existence of a bilateral coordination mechanism for compliance with forestry law (BCM), as is the case in China. Sources: [58–64].

The following equation tests the link between the independent variables and the dependent variable:

$$IT_{it} = v_{it} + \alpha_1\,CPI_{it} - \alpha_2\,TCL_{it} + \alpha_3\,CFH_{it} + \alpha_4\,TD_{it} + \alpha_5\,TS_{it} - \alpha_6\,D_{it} + \alpha_7\,Ag_{it} + e_{it} \qquad (1)$$

where:

| | |
|---|---|
| i | country |
| t | year |
| $IT_{it}$ | EU imports |
| $V_{it}$ | other exogenous variables not included in the model |
| $\alpha$ | estimated coefficients |
| $CPI_{it}$ | Transparency Index |
| $TCL_{it}$ | tree cover loss |
| $CFH_{it}$ | certified forest hectares |
| $TD_{it}$ | EU timber demand |
| $TS_{it}$ | timber supply |
| $D_{it}$ | distance |
| $Ag_{it}$ | Voluntary Partnership Agreement or similar |
| $e_{it}$ | error term |

The use of panel data methodology is appropriate due to the inclusion of time periods and the probable presence of unobserved individual effects. It captures the influence of variables that are not measured but that may explain the variation between countries. The variables are presented in logarithms, and a sequence of econometric models is formulated until the optimum is identified. Panel data techniques involve a combination of cross-sectional and time-series analyses, focusing on specific units under analysis and enabling observations to be followed over time, controlling for

unobservable individual heterogeneity. Indeed, the countries in the sample are heterogeneous due to their geographical, historical, political and economic differences; these are specific factors that may be affecting EU imports of timber and timber products but that are difficult to measure. Other advantages provided by this technique are that it reduces the collinearity between variables, provides more degrees of freedom and more efficiency, is better able to study the dynamics of adjustment, allows the identification and measurement of effects that time-series or cross-sectional analyses do not detect, enables more complex models to be built and tested and eliminates or reduces information aggregation bias in the results [65]. However, among the drawbacks, it is worth pointing out problems in the design and sourcing of data, limitations caused by short time-series and cross-sectional dependence.

## 4. Results

### 4.1. Descriptive Analysis

First, a descriptive analysis of the variables is carried out (Table 2). There is a high degree of dispersion in the main variable—imports of timber and timber products—as well as in some of the independent variables, such as the certified forest hectares (CFH) and the timber supply (TS). For the latter, a value of 0 is recorded for certain countries such as Albania, the Democratic Republic of the Congo, Côte d'Ivoire, Equatorial Guinea, Morocco, Myanmar and the Central African Republic. The rest of the variables show less variation—in particular, the EU timber demand (TD).

**Table 2.** Descriptive statistics of EU imports of timber and timber products and the independent variables between 2012 and 2017.

| Variables | Obs | Mean | Std. Dev. | Coef. Var. % | Min | Max |
|---|---|---|---|---|---|---|
| IT | 228 | 233,223.90 | 374,530.90 | 1.60 | 1292.10 | 1,900,000.00 |
| CPI | 228 | 43.94 | 20.14 | 0.45 | 15.00 | 91.00 |
| TCL | 228 | 0.53 | 0.41 | 0.76 | 0.05 | 1.78 |
| CFH | 228 | 8,189,094.00 | $2.68 \times 10^7$ | 3.27 | 0.00 | $1.69 \times 10^8$ |
| TD | 228 | 685,725.80 | 30,739.74 | 0.04 | 647,466.60 | 736,690.00 |
| TS | 228 | 24,778.74 | 68,347.08 | 2.75 | 39.00 | 418,912.00 |
| D | 228 | 6982.01 | 4180.36 | 0.59 | 670.00 | 18,711.00 |
| Ag | 228 | 0.73 | 0.44 | 0.60 | 0.00 | 1.00 |

Source: Own elaboration from [3,55–61].

The analysis of the independent variable confirms that timber trade is of major importance in the EU. In 2012, following the slowdown in the world timber market due to the economic crisis that began at the end of 2008, purchases of timber and timber products on the international market by all EU countries totalled $37.87 billion, representing 37.01% of the total, as shown in Table 3. China, with imports valued at $14.69 billion and a share of 14.35%, is in second position, followed by the USA (11.54%) and Japan (10.51%). Imports by the rest of the countries amounted to less than $2.5 billion, accounting for 26.59% of the total.

**Table 3.** Distribution of world imports of timber and timber products by country groups and countries in 2012 and 2017 and variations over the period 2012–2017 (%).

| Countries | 2012 | | 2017 | | Var 2012–2017 |
|---|---|---|---|---|---|
| | Billions of $ | % | Billions of $ | % | |
| EU | 37.87 | 37.01 | 43.24 | 35.63 | 14.17 |
| China | 14.69 | 14.35 | 22.76 | 18.75 | 54.96 |
| USA | 11.81 | 11.54 | 18.49 | 15.23 | 56.53 |
| Japan | 10.76 | 10.51 | 9,15 | 7.54 | −14.93 |
| Republic of Korea | 2.35 | 2.30 | 3.16 | 2.61 | 34.46 |
| Canada | 2.84 | 2.77 | 2.72 | 2.24 | −4.16 |

| | | | | | |
|---|---|---|---|---|---|
| India | 2.56 | 2.50 | 2.14 | 1.77 | −16.37 |
| Switzerland | 1.74 | 1.70 | 1.71 | 1.41 | −1.50 |
| Australia | 1.33 | 1.30 | 1.64 | 1.35 | 23.40 |
| Mexico | 1.30 | 1.27 | 1.50 | 1.23 | 15.59 |
| Norway | 1.54 | 1.50 | 1.45 | 1.19 | −5.84 |
| Egypt | 1.66 | 1.62 | 1.30 | 1.07 | −21.29 |
| Turkey | 1.55 | 1.52 | 1.05 | 0.86 | −32.60 |
| Rest of the world | 10.35 | 10.11 | 11.05 | 10.80 | 6.84 |
| Total | 102.34 | 100.00 | 121.36 | 100.00 | 18.59 |

Source: [3].

During the period analysed, the data in Table 3 reveals very uneven dynamics in the imports of timber and timber products by the different countries and the EU as a whole. In particular, the value of the EU's purchases increases by 14.17%, although its share of the world total drops to 35.63%. Compared to this dynamic, China records an annual rate of change of 9.15%, reaching an import volume of almost $3 billion, 54.96% higher than the value recorded five years earlier. Similarly, the USA experiences a marked increase in the value of its purchases on the international market, becoming the destination country for 15.23% of the total. Notable, albeit smaller, increases are registered in the Republic of Korea, Australia and Mexico. In the opposite direction, Japan, Canada, India, Switzerland, Norway, Egypt and Turkey record lower import volumes in 2017 than in 2012.

At the same time, the origin of imports from outside the EU shows very little diversification, as can be seen in Table 4. More than two-fifths of all imports come from three major markets: Russia, China and the United States. These are followed by 15 very heterogeneous economies located on four continents—Europe, Africa, America and Asia—which together account for 50% of the total.

**Table 4.** Distribution by country of origin of timber and timber product imports from outside the EU-28 and cumulative average growth rate, 2012–2017 (%).

| Countries | 2012 | 2013 | 2014 | 2015 | 2016 | 2017 | Var 2012–2017 |
|---|---|---|---|---|---|---|---|
| Russia | 17.58 | 18.41 | 18.24 | 16.29 | 17.12 | 18.26 | 6.03 |
| China [2] | 15.81 | 14.10 | 13.84 | 14.00 | 12.45 | 12.10 | −0.25 |
| USA | 10.18 | 11.35 | 13.28 | 15.22 | 15.13 | 15.03 | 13.77 |
| Ukraine | 5.98 | 6.33 | 7.08 | 7.34 | 7.96 | 7.76 | 10.87 |
| Norway | 4.77 | 5.39 | 5.40 | 5.30 | 5.09 | 5.43 | 8.01 |
| Brazil | 5.77 | 5.06 | 5.12 | 5.39 | 4.41 | 4.54 | 0.27 |
| Canada | 4.94 | 6.00 | 4.30 | 4.30 | 4.33 | 4.07 | 1.23 |
| Belarus | 2.92 | 3.19 | 4.09 | 4.17 | 5.13 | 6.70 | 24.20 |
| Indonesia [1] | 4.50 | 3.87 | 3.79 | 4.25 | 4.10 | 4.00 | 2.77 |
| Switzerland | 4.74 | 4.55 | 4.38 | 3.82 | 3.66 | 3.52 | −0.85 |
| Malaysia | 4.48 | 3.88 | 3.60 | 3.51 | 3.09 | 2.94 | −3.31 |
| Cameroon [1] | 3.52 | 2.95 | 2.68 | 2.71 | 3.03 | 2.33 | −3.07 |
| Bosnia Herzegovina | 2.07 | 2.55 | 2.57 | 2.40 | 2.60 | 2.59 | 10.05 |
| Gabon | 1.79 | 1.88 | 1.67 | 1.67 | 1.96 | 1.67 | 3.77 |
| Chile | 1.14 | 0.90 | 1.07 | 1.09 | 1.01 | 1.00 | 2.53 |
| Serbia | 0.88 | 1.05 | 1.13 | 0.96 | 1.03 | 1.05 | 9.04 |
| Uruguay | 1.24 | 1.24 | 0.85 | 0.85 | 1.05 | 0.84 | −2.65 |
| Congo [1] | 0.68 | 0.83 | 0.75 | 0.74 | 0.83 | 0.70 | 5.67 |
| Ghana [1] | 0.52 | 0.44 | 0.40 | 0.31 | 0.30 | 0.26 | −8.29 |
| Vietnam [1] | 0.28 | 0.30 | 0.25 | 0.26 | 0.24 | 0.23 | 1.32 |
| Central African R [1] | 0.12 | 0.09 | 0.07 | 0.12 | 0.14 | 0.07 | −5.73 |
| Liberia [1] | 0.14 | 0.06 | 0.03 | 0.03 | 0.02 | 0.01 | −34.79 |
| Rest of the world | 5.94 | 5.58 | 5.39 | 5.26 | 5.31 | 4.89 | 1.22 |
| Total | 100.00 | 100.00 | 100.00 | 100.00 | 100.00 | 100.00 | 5.23 |

[1] Countries that have signed Voluntary Partnership Agreements (VPAs) with the EU. [2] Country that has a bilateral cooperation mechanism with the EU-28 on forest law compliance since 2009. Source: [3].

The rest of the non-EU trade partners account for a small share: less than 1% in all years. We can observe very uneven dynamics in the markets for suppliers of timber and timber products to the EU between 2012 and 2017; while imports from Belarus, Myanmar and New Zealand present annual average growth rates of over 20%, other countries register a decline in the value of their sales, as is the case with Cameroon, Ghana, the Central African Republic and Liberia, all four of which have signed a VPA.

*4.2. Panel Data Analysis*

Table 5 shows the results of the panel data estimations. The model contains information from 38 countries, with a total of 228 observations, for the time period 2012–2017. The panel data methodology is chosen for the analysis in order to capture the influence of unobserved variables that can explain the variation between countries. A series of econometric models are formulated until the optimum model is reached:

(a) After performing an exploratory analysis of the data, the procedure begins by estimating the model with pooled data and then with random effects. The Lagrangian Multiplier Test is used to compare the two models. To choose between the pooled data model or the random effects model, the Lagrangian Multiplier Test for random effects is implemented under the null hypothesis that the variance of the individual unobservable effect $V_{it}$ is zero. The value of $p = 0$ obtained indicates that the null hypothesis must be rejected (Prob > chi$^2$ = 0.0000) and that, therefore, the random effects approach should be used. There is, therefore, evidence of significant differences between countries.

(b) Once the pooled data approach is discarded, a choice must be made between fixed effects and random effects. To do so, the Hausman test is performed under the null hypothesis that the two estimators do not differ significantly. If this is rejected, the fixed-effects approach is preferable; otherwise, random effects should be chosen due to their greater efficiency. The two models are compared without time dummies, as it is confirmed that the time effect is not relevant in either the random effects model or the fixed-effects model. The results of the Hausman test indicate that the null hypothesis is rejected; therefore, fixed effects should be chosen.

(c) A series of tests are then run on the fixed-effects estimator to correct for problems (Table 5). The Pesaran and Frees tests are used to check for cross-sectional dependence in the errors, with the rejection of the null hypothesis indicating dependence. In both cases, the null hypothesis is rejected, confirming the existence of this problem. Another problem is the presence of autocorrelation or first-order serial correlation. The autocorrelation is verified with the Wooldridge test under the null hypothesis that there is no autocorrelation. The rejection of the hypothesis indicates that there is autocorrelation that must be corrected. The results (Table 5) confirm the need to correct for both these problems, so a first-order autoregressive term (AR-1) is introduced into the fixed-effects model, referred to as the AR-1 model. In addition, the presence of heteroskedasticity in the data is confirmed by means of the modified Wald test under the null hypothesis that there is no heteroskedasticity in the fixed-effects model. The null hypothesis is rejected in this study, confirming the presence of heteroskedasticity. All these problems, which are fairly common in social sciences research, can be solved by the Prais-Winsten transformation through panel-corrected standard errors (PCSE), which is the one recommended for fixed effects [66]. It is assumed that AR-1 autocorrelation is present within the panels and that the coefficients of this process are specific to each group. It is also assumed that the residuals are by default heteroskedastic. Therefore, the optimum model is the Prais-Winsten transformation correcting the standard errors, with the time effect not proving to be significant.

The data in Table 5 confirm the relationship between the dependent variable and independent variables in all cases, albeit with different levels of significance; the only exception is the variable certified forest hectares, which is not significant. Therefore, hypothesis 3 is not confirmed. At 1% significance and with the expected sign, the Transparency Index, the percentage of tree cover loss, the EU construction sector (as a proxy for timber demand) and the distance between the EU and the exporting country are found to be determining factors, confirming the direct link (association) between the activity of the construction sector and EU timber imports; in addition, the inverse relationship between imports and the distance from the exporting country is confirmed. The timber supply and the trade agreements (at 5%) are found to have a weaker influence on the dependent variable and with the opposite sign to the expected one, meaning that the probability of the EU importing is not influenced by the production capacity of the supplier country or the signing of agreements with the EU.

**Table 5.** Panel data estimates.

| Dependent Variable: Import Item | PCSE | Tests | Tests Results |
|---|---|---|---|
| CPI | 0.034 *** | | |
| | (0.009) | Hausman | $Chi^2$ (6) = 54.23; prob > $chi^2$ = 0.0000 |
| TCL | −0.756 *** | | |
| | (0.154) | | |
| CFH | −0.025 | Pesaran (*p*) and Frees (F) | *p* = 5.912; Pr = 0.0000 |
| | (0.020) | | F = 0.899 |
| TD | 2.815 *** | | |
| | (0.378) | Wooldridge | F(1, 37) = 58.275; prob > F = 0.0000 |
| TS | −0.130 ** | | |
| | (0.064) | | |
| D | −3.716 *** | Wald−modified | $Chi^2$(38) = 4819.54; prob > 0.0000 |
| | (0.660) | | |
| Ag | −0.387 ** | | |
| | (0.176) | | |
| Observations | 228 | | |
| $R^2$ | 0.998 | | |
| Number of countries | 38 | | |

*** Significant at 1% and ** at 5%. Standard errors in parentheses. Source: own elaboration from [3,58–64]. PCSE: panel-corrected standard errors.

The panel data analysis shows that the transparency levels, the tree cover loss, the demand, (linked to the construction sector) and the distance from the supplier country are the variables that have the biggest influence.

## 5. Discussion

The analysis carried out confirms that the EU is a major player in world trade in timber and timber products, as reported by the EU Timber Regulation Biennial Implementation Report 2015–2017 [67]. Between 2012 and 2017, EU countries purchased timber and timber products on the international market worth $242.97 billion, according to the UN COMTRADE database. During these years, these flow registered an upwards trend, confirming the EU's position as the main destination for world imports of these products. At the same time, China and the USA recorded more marked increases in their purchases. Some studies pointed to less stringent or inexistent requirements and

regulations as the reason for trade shifting towards China, which became the destination country for 18.75% of world imports of timber and timber products [21,23].

The origin of imports shows very little diversity. In 2017, seven countries (Russia, China, the USA, Ukraine, Norway, Brazil, Canada, Belarus and Indonesia) provided the EU with more than three-quarters of its total imports. Three of these countries are particularly notable in this regard: Russia (providing 18.26% of the total), the USA (15.03%) and China (12.10%). In these years, China's share of timber supplies to the EU market dropped by more than three percentage points, registering lower sales in 2017 than in 2012. Conversely, there was an increase in those coming from geographically closer supplier countries, such as Russia, Belarus or the Ukraine. At the same time, the USA, which relies on the Lacey Act to combat illegal timber, is seeing very significant increases in its sales to the EU market. Countries that have signed a VPA with the EU represent a small share of timber supply to the EU, with all these countries accounting for less than 1.00% each, except Indonesia (4.00%) and Cameroon (2.33%). Moreover, with the exception of the Congo, these markets have not registered any significant increases in their exports to the EU between 2012 and 2017. On the contrary, they register an average annual change of less than 5.00%. Four countries have even registered a decline, which may indicate a substitution effect in favour of other markets [8]. The share of total EU timber imports for the group of countries with a VPA dropped from 9.64% in 2012 to 7.52% in 2017. In parallel, China's relative importance as a supplier country decreased by more than five percentage points, to 19.7% in 2017. Despite the objectives included in each of the VPAs signed by the EU, these data call into question the ability of agreements on the legality of timber to facilitate, from the outset, access to the European market for the supplied products. Such an outcome could be inferred from the postulates of the proposed gravity model outlined in Section Two. However, the results of this process are specific to individual countries [30].

According to the results of the analysis performed, hypotheses 1 and 2 are confirmed. Conversely, hypothesis 3 is not confirmed: the variable certified forest hectares is not significant. Table 6 presents the main results according to the proposed hypotheses. The variables Transparency Index and tree cover loss are found to be very important in determining EU timber imports. The transparency of the supplying countries' institutions shows a strongly positive and significant association with EU imports of timber. This confirms hypothesis 1, as reported by the European Commission [25] and Borsky et al. [26], countries with highly transparent government institutions that have registered a rise in EU imports of timber in recent years. The variable tree cover loss is also significant, having an inverse relationship with the main variable, confirming hypothesis 2, in-line with the studies by with Park [28] and Youn and Vadell et al. [29]. The percentage of tree cover loss may be indicative of the type of governance and national policy applied to tackle deforestation and forest degradation, such that a high percentage of tree cover loss will have a negative effect on EU timber imports. These two factors—the Transparency Index and tree cover loss—influence the probability of importing from non-EU countries. The level of transparency of the institutions in the supplier country helps ensure the legality of the exported products, and the percentage of tree cover loss is an indicator of sustainable forestry practices. Conversely, hypothesis 3 is not confirmed. The results indicate that the number of certified hectares in the exporting country does not affect the total EU imports of timber and timber products. The costs assumed by the company when opting for an FSC or PEFC certification system compared to cheaper processes linked to the implementation of national forest management plans could explain why exporting companies might opt for legality but not for certification. In fact, as Van Kooten, Nelson and Vertinsky report [68], many countries have national certification standards, as is the case with Canada or Malaysia. Contrary to expectations, agreements on the legality of imported timber that the EU-28 has signed with third countries have a negative influence in most cases. These results are in-line with the study of Brusselaers and Buysse regarding findings from Cameroon in 2011 [12].

**Table 6.** Hypotheses and main results. EUTR: EU Timber Regulation, FSC: Forest Stewardship Council and PEFC: Program for the Endorsement of Forest Certification.

| Hypotheses | Main Results |
| --- | --- |
| H1. The requirements established by the EUTR for the import of legal timber have promoted EU imports of timber from countries that have highly transparent government institutions. | H1 confirmed: Transparency Index is significant at 1%, having the expected sign (a direct association with the variable timber imports). |
| H2. Greater tree cover loss in the exporting country negatively affects EU timber imports. | H2 confirmed: tree cover loss is significant at 1%, having the expected sign (inverse relationship with the variable timber imports). |
| H3. There is a positive relationship between the number of forest hectares with FSC or PEFC certification in exporting countries and EU timber imports. | H3 not confirmed: certified forest hectares is not significant. |

Other variables that have a notable influence on EU imports of timber and timber products from the non-EU market are the timber demand and the distance of the supplier country, in-line with Hurmekoski et al. [16], Paluš et al. [17], Manninen [55], O'Brien and Bringezu [56] and Morland et al. [57], confirming the direct link between the activity of the construction sector and EU timber imports and the inverse relationship between imports and the distance from the exporting country. These findings should be taken into account when designing actions affecting the private sector, especially the construction sector, which seeks to ensure the effectiveness of the FLEGT Action Plan and, ultimately, improve forest governance and sustainable forest management.

## 6. Conclusions

The EUTR prohibits the placing of illegal timber and timber products on the EU-28 market to help slow down the process of degradation and deforestation of forests. The aim of the regulation is to contribute to improving forest management in producing countries, as stated in the regulation itself. In the preceding pages, we analysed the behaviour of EU imports of these products and, specifically, EU-28 imports of those products included in Chapter 44 of the EUTR between 2012 and 2017, studying the determinants of their evolution. The results obtained show that the EU regularly imports timber and timber products from third countries, and the policies it adopts influence the legality of the practices of producers and suppliers in the international market. EU imports are directly influenced by the level of transparency in supplier economies, as well as the behaviour of the construction sector, which determines the demand. Conversely, poor environmental performance, as reflected in the tree cover loss, and an exporting country's distance from Brussels negatively influence EU imports from the international market. In the short and medium-term, the agreements on the legality of imported timber (the VPA agreements and the one signed with China), which reflect the parties' joint commitment to legal logging and the associated timber trade, are not found to be driving a greater volume of exports to EU countries; on the contrary, an adverse effect is observed.

The arguments presented call into question the ability of the FLEGT Plan to positively influence, in the short and medium-term, exports of timber and timber products from the VPA countries to the EU market. In most cases, economies that have a VPA report lower income from their exports of timber and timber products to the EU, which may affect the willingness of national governments to initiate VPA negotiations. In turn, this may impact the effectiveness of the FLEGT Action Plan aimed at improving forest management. In addition, both importers and exporters raise issues, such as weak law enforcement, insufficient guidance from regulatory and implementing authorities and increased bureaucracy in both EU and partner countries [22,69]. This detracts from the effectiveness of the EU Timber Regulation as an instrument to support the legality of trade flows. More diligence and clarity is required in the development of the agreements to prevent situations of uncertainty that negatively

affect export activity, as well as financing to help the partner countries deal with the start-up costs of the process, as has happened in the case of Ghana, Indonesia and Laos [70]. In-line with the proposal in the conclusion of the Evaluation of the EU FLEGT Action Plan 2004–2014 [71], there is a need to focus on international coalitions in order to tackle the illegal logging and timber trade worldwide [24]. This would help to meet the commitments identified in the United Nations Strategic Plan for Forests 2017–2030.

This research presents an exploratory analysis of the evolution of EU countries' imports of timber and timber products from third countries. The results obtained should be assessed, taking into account the fact that other factors not considered in this study may exert an influence. In this regard, we believe further complementary research is needed. It would be worth analysing the specific factors affecting imports broken down by type of product, as well as the flows from different partner countries and, in particular, those occurring within the EU. It is also necessary to clarify patterns in the trade flows of EU countries with substantial trade activity, such as Romania and Germany, and to analyse whether the variables studied here have the same impact on the imports of the different national economies of the EU.

**Author Contributions:** Conceptualisation, E.M.-P., C.M.-A. and Á.A.C.-C.; methodology, C.M.-A. and L.G.-V.; validation, E.M.-P. and L.G.-V.; formal analysis, E.M.-P.; investigation, E.M.-P. and C.M.-A.; resources, E.M.-P., C.M.-A. and L.G.-V.; data curation, C.M.-A.; writing—original draft preparation, E.M.-P., Á.A.C.-C. and C.M.-A.; writing—review and editing, E.M.-P. and L.G.-V.; visualisation, E.M.-P.; supervision, E.M.-P. and project administration, E.M.-P. All authors have read and agreed to the published version of the manuscript.

**Funding:** This research received no external funding.

**Conflicts of Interest:** The authors declare no conflicts of interest.

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
