# Peer review of "Transparency Index of the Supplying Countries’ Institutions and Tree Cover Loss: Determining Factors of EU Timber Imports?"

_forests, doi:10.3390/f11091009_

Round 1
Reviewer 1 Report
Thank you very much for this interesting article. It was a pleasure reading it. It present valuable data on the EUTR implementation.
The article analyses the evolution of EU timber imports from third countries, with a special emphasis on the variables affecting this trade flow.
The article is well written, the introduction sufficiently outlines the problem, formulates research questions and identifies the research gap in the effects of EUTR on EU timber trade. As the authors claim they seek to verify the possible link between EU imports of timber and timber products from third countries and selected economic and institutional variables. To analyse this issue, they formulate seven research questions. Based on this questions 3 hypothesis are formulated. Their formulation is sound and based on literature review.
Methodology is well described.
Results are presented in a clear way. The only reservation I have concerns the presentation of the answers to the RQ and hypothesis testing. It would be beneficial for the readers and clarity of presented results to add a table summarizing the research questions, hypothesis and main results. It can be read in the text but a summarizing table would even better illustrate the obtained results.
I would have expected a broader discussion of the selected variables and their confrontation with similar research conducted.
Conclusions are clear and supported by the data.
I find the title is a little bit misleading, because the results show that not only these two factors determine the EU timber import. I would suggest to change the title.
Author Response
We sincerely appreciate all the work done with the comments, proposals and indications made, which have been taken into account in the review of this paper, and have allowed its improvement. All the changes are in red.
Thank you very much for this interesting article. It was a pleasure reading it. It present valuable data on the EUTR implementation.
The article analyses the evolution of EU timber imports from third countries, with a special emphasis on the variables affecting this trade flow.
The article is well written, the introduction sufficiently outlines the problem, formulates research questions and identifies the research gap in the effects of EUTR on EU timber trade. As the authors claim they seek to verify the possible link between EU imports of timber and timber products from third countries and selected economic and institutional variables. To analyse this issue, they formulate seven research questions. Based on this questions 3 hypothesis are formulated. Their formulation is sound and based on literature review.
Methodology is well described.
Results are presented in a clear way. The only reservation I have concerns the presentation of the answers to the RQ and hypothesis testing. It would be beneficial for the readers and clarity of presented results to add a table summarizing the research questions, hypothesis and main results. It can be read in the text but a summarizing table would even better illustrate the obtained results.
ANSWER: A table has been introduced summarizing the results of the hypotheses. The research questions, at the request of the other 2 reviewers, have been removed.
I would have expected a broader discussion of the selected variables and their confrontation with similar research conducted.
ANSWER: It has been expanded and confronted with literature.
Conclusions are clear and supported by the data.
I find the title is a little bit misleading, because the results show that not only these two factors determine the EU timber import. I would suggest to change the title.
ANSWER: The title, also the abstract and the text have been modified a little, placing more emphasis on these variables, which are the main ones.

Reviewer 2 Report
The manuscript presents an analysis of EU timber imports after the implementation of the EUTR. The topic of the manuscript is actual and fits with the scope of the journal. Utilisation of panel data methodology is discussable; however in my opinion applying in this case is correct.
The following recommendations are intended to be seen as a help to the authors in order to improve the manuscript.
I found merit in the detail and comprehensive study, but also had some minor concerns. The abstract is quite general and I miss a more detail description of results.
Introduction must be improved by particular literature review. (e.g. Line 86 or Line 96, Line 107). Many similar national research are provided on the forestry industry ( e.g. Paluš, H. et al. (2018). "Determinants of sawnwood consumption in Slovakia," BioRes. 13(2), 3615-3626. etc.)
I am not really understand the reason of questions (Line 70-81). The authors did not answer to them in the Discussion or Conclusions) I suggest to cancel them.
The authors must improve their English (e.g. incorrect formulation of phrases, applying of articles or titles – word like “chapters” Line 88 – 94, etc.) and formal errors (e.g. authors meny times mentioned literature review without particular authors, or mistakes like: Line 234 “Table 2” it should not be 1?, Line 265 why “Lastly ? Vit should be on the first position after time and country “i”(shoul be added). For next variables repeat country and time is not necessary to repeat.
Line 332 - 333 Authors should explain in detail how optimum model has been reached.
Line 341 352 – descriptions of tests are very general. Tests must be described in detail with calculated values.
The chapter “Discussion” has descriptive and quite general character. Discussion is focused on description of statistical data; however authors should in more detail focus on model. From this point of view they have to compare and highlight their new knowledge of their research.
In Discussion they must add comparison with similar research. These research should be mentioned in Introduction.
I hope, these remarks will be useful for authors.
Author Response
REVIEWER 2
We sincerely appreciate all the work done with the comments, proposals and indications made, which have been taken into account in the review of this paper, and have allowed its improvement. All the changes are in red.
The manuscript presents an analysis of EU timber imports after the implementation of the EUTR. The topic of the manuscript is actual and fits with the scope of the journal. Utilisation of panel data methodology is discussable; however in my opinion applying in this case is correct.
The following recommendations are intended to be seen as a help to the authors in order to improve the manuscript.
I found merit in the detail and comprehensive study, but also had some minor concerns. The abstract is quite general and I miss a more detail description of results.
ANSWER: Abstract changed, specific data added on variables of importance in the study.
Introduction must be improved by particular literature review. (e.g. Line 86 or Line 96, Line 107). Many similar national research are provided on the forestry industry ( e.g. Paluš, H. et al. (2018). "Determinants of sawnwood consumption in Slovakia," BioRes. 13(2), 3615-3626. etc.)
ANSWER: Bibliography added. In addition, the introduction and bibliographic analysis (sections 1 and 2) have been modified, adding the names of the authors studied.
I am not really understand the reason of questions (Line 70-81). The authors did not answer to them in the Discussion or Conclusions) I suggest to cancel them.
ANSWER: They have been eliminated.
The authors must improve their English (e.g. incorrect formulation of phrases, applying of articles or titles – word like “chapters” Line 88 – 94, etc.) and formal errors (e.g. authors meny times mentioned literature review without particular authors, or mistakes like: Line 234 “Table 2” it should not be 1?, Line 265 why “Lastly ? Vit should be on the first position after time and country “i”(shoul be added). For next variables repeat country and time is not necessary to repeat.
ANSWER: The text has been extensively corrected following the reviewer's instructions. Authors have been added in the introduction and in the bibliographic analysis (sections 1 and 2).
The word “Chapter” is the one used by the Combined Nomenclature: https://ec.europa.eu/taxation_customs/business/calculation-customs-duties/what-is-common-customs-tariff/combined-nomenclature_en
Line 332 - 333 Authors should explain in detail how optimum model has been reached.
Line 341 352 – descriptions of tests are very general. Tests must be described in detail with calculated values.
ANSWER: It has been explained in detail, organizing it with numbering to make it clearer and adding data in table 5.
The chapter “Discussion” has descriptive and quite general character. Discussion is focused on description of statistical data; however authors should in more detail focus on model. From this point of view they have to compare and highlight their new knowledge of their research.
In Discussion they must add comparison with similar research. These research should be mentioned in Introduction.
ANSWER: The discussion section has been expanded and contrasted with the literature analyzed. The similar research are mentioned in the introduction and bibliographic analysis sections (sections 1 and 2).
I hope, these remarks will be useful for authors.
